# Investigation on Deformation Behavior in the Surface of Metal Foil with Ultrasonic Vibration-Assisted Micro-Forging

**DOI:** 10.3390/ma15051907

**Published:** 2022-03-04

**Authors:** Zidong Yin, Ming Yang

**Affiliations:** Graduate School of System Design, Tokyo Metropolitan University, 6-6, Asahigaoka, Hino-shi 191-0065, Japan

**Keywords:** surface finishing, surface deformation, micro-forging, ultrasonic vibration-assisted

## Abstract

Excitation of the acoustic field, leading to the Blaha effect, significantly affects the plasticity of a material. In the micro-forming field, the so-called impact effect is found to generate a larger amount of dislocation and produce greater plastic deformation than acoustic softening. In this study, the mechanism of deformation in the surface of the material with ultrasonic vibration assistance was investigated and compared with that in the bulk. Forging tests using a newly developed ultrasonic vibrator were carried out on pure Cu foils with various process conditions. The longitudinal vibration frequency of the ultrasonic transducer was 60 ± 2 kHz, and the vibration amplitude was in an adjustable range of 0~6 μm. Forging tests were carried out at different amplitudes. The result shows that acoustic softening and the impact effect could be separated by an oscilloscope in the micro-forging system. The difference in deformation on the surface asperity caused by acoustic softening and the impact effect is discussed. Compared to acoustic softening, which has a limited effect on the deformation of the surface asperity, the impact effect could create more plastic deformation on the surface asperity. Therefore, the reduction in the surface roughness would increase after the impact effect occurs. In addition, to confirm the mechanism of acoustic softening and the impact effect, the microstructural evolution of specimens, at the surface scale and inner scale, was investigated by electron backscatter diffraction (EBSD). It was found that acoustic softening could create more grain refinement, and with the amplitude increasing, the impact effect would oppositely cause the surface grains to grow. In this study, the mechanism of how the impact effect and acoustic softening affect the deformation behavior of the surface asperity was investigated.

## 1. Introduction

Due to the increasing demand for micro technical products, such as micro-electro-mechanical systems (MEMS), the process of micro-forming has been widely used to produce miniature metallic parts. Nevertheless, some problems such as the so-called size effect will be caused by scaling down the ratio of roughness until its dimensions decrease, increased forming stress [1,2], and lower forming accuracy in the process behavior [3]. In the past decade, plenty of researchers have tried to use different, additional energies to increase surface properties and form accuracy. Compared with other methods, ultrasonic vibration assistance, as a relatively more efficient processing method with lower energy loss in the field of micro-forming, has garnered increased attention from both academic and industrial fields.

In 1955, Austrian scientists Blaha and Langenecker found that ultrasonic vibration could reduce the yield stress significantly in the tensile test of a zinc single crystal, for the first time [4]. Researchers have devoted much effort to applying ultrasonic vibration to micro-manufacturing devices. Jimma et al. (1998) designed an experimental apparatus applying ultrasonic vibration to a deep drawing process, which increased the limiting drawing ratio in both rolled steel and 304 stainless steel [5]. Weidong Zhai and li Yanle (2019) studied the ultrasonic-assisted incremental sheet forming process, and the result showed that the forming force was considerably reduced, and the bearing capacity could be improved. Both of these effects were more obvious at the step-down size or in a thin material sheet [6]. Chunju Wang (2017) presented a copper foil’s mechanical properties with ultrasonic vibration tensile testing, considering the size effect in particular. The result showed that the flow stress was decreased by ultrasonic vibration with different geometry sizes, and more significantly when the size was scaled down [7].

Previous research has established that ultrasonic vibration affects materials in two aspects: volume and surface. The reported effect on the volume is that when ultrasonic vibration is applied to the tool, a reduction in the yield stress of the material will occur to a certain degree. As for the effect on the surface, Bai and Yang (2016) obtained a better surface finish by using an innovative method to apply ultrasonic vibration; surface asperities’ deformation and the reduction in surface roughness were linear to the ultrasonic vibration amplitude, and static stress was produced [8]. However, the fundamental mechanisms of how ultrasonic vibration affects materials are still controversial. There are two generally accepted explanations at the macro scale: stress superposition and acoustic softening. Stress superposition is only an apparently average stress reduction method, with periodic loading and unloading; this is explained as the framework of the elastic–plastic behavior of the material under uniaxial deformation (Kirchner et al., 1985), and it is thought that no microstructural changes are induced by stress superposition [9]. On the other hand, acoustic softening works by decreasing the real flowing stress and changing the material properties or microstructure (Langenecker. 1966) [10]. Ultrasonic vibration offers energy for dislocation, which can be part of the required activation energy for dislocations to overcome lattice obstacles so that the mobility of dislocations is enhanced. J. Hu et al. (2020) showed that when acoustic softening occurs, compared to the condition without ultrasonic vibration, the dislocations/low-angle grain boundaries distribute randomly in grains, which results in the motion of dislocations/low-angle grain boundaries being improved by acoustic softening [11]. At the micro level, this is expressed as a stress reduction. Furthermore, apart from stress superposition and acoustic softening, dynamic impact is reported as playing an important role in ultrasonic-assisted compression tests. In ultrasonic vibration-assisted compression tests, the dynamic effect occurs whether the vibrated punch is able to detach from the specimen periodically or not. The deformation caused by the ultrasonic dynamic impact effect is gradually accumulated by the repeated impact from the punch and the bottom die. The grain size is refined, and low-angle grain boundaries appear on the top side and in the center of the material.

Indeed, some researchers have studied how ultrasonic vibration affects the surface of a material. It has been demonstrated that, at the same vibration time, the forming work is increased with frequency, and that a better surface finish and a higher degree of micro-hardness can be gained with a higher frequency (Yang and Bai 2013) [12]. The impact effect will gradually become more dominant when the amplitude of ultrasonic vibration increases, which will lead to a further reduction in the forming force and a greater change in the microstructure (J. Hu et al.) [13,14]. However, when Yang and Bai [12] investigated the ultrasonic effect’s influence on the surface of the material, the ultrasonic energy, which consists of frequency and amplitude, was not large enough to be considered as ultrasonic, and it was also difficult to discuss acoustic softening and the impact effect separately during the micro-forging without a dynamic force test system, not to mention comparing the different changes in microstructure induced by ultrasonic vibration in the surface and inner material. Although J. Hu [13] obtained and discussed the influence of the impact effect on the reduction in forming stress in his compression test, the deformation of the surface asperity was too large to be easily observed.

In this study, to investigate the deformation behavior in the surface under ultrasonic vibration at the micro scale, considering the dynamic impact effect and acoustic effect, a novel ultrasonic-assisted micro-forging system was used. Micro-forging tests of commercial copper were carried out in different conditions. The effects of different amplitudes on surface topographies, the reduction in the surface roughness, and the deformation of surface asperities were investigated. Additionally, in order to investigate the difference in microstructure between the top side and deep side of the material during micro-forging under diverse amplitudes, EBSD (electron backscatter diffraction) was used to gain a better understanding of the underlying mechanisms.

## 2. Experimental Setup

### 2.1. Ultrasonic-Assisted Micro-Forging Test System

In this study, a novel ultrasonic-assisted micro-forging test system including an upper die assembly, a lower die, and an ultrasonic vibrator was set up, as shown in Figure 1. The ultrasonic vibrator contained two horizontal ultrasonic transducers, a horn at a resonance frequency of 60 kHz, and a punch for forging.

Figure 2 shows the integrated structure of the ultrasonic vibrator.

Two horizontal transducers generated ultrasonic vibration, originally in the same phase. Then, the punch received the vibration, which was amplified and transformed in a vertical direction by a specially designed horn. Finally, the maximum amplitude of 6 μm on the punch tip was measured by a laser displacement meter (LC-2400, KEYENCE, Tokyo, Japan). The input electrical signal for the ultrasonic transducers was a sine wave, with a frequency of 60 kHz, from an ultrasonic signal generator. The die and punch were made from tool steel (SKD11, ADD.Q, Kanagawa, Japan), and the surfaces were fine grinded with a heat treatment, keeping the surface roughness at Ra = 0.1 μm. The punch surface was re-ground to keep the surface clear and to ensure no residual specimen was left after each experiment. A stress load cell is located on the ultrasonic vibrator which measures the pre-load, and since it connects to the strain amplifier, the strain can also be obtained. In order to prevent warping of the copper chip during processing, a plate with four screws distributed in four corners and a through-hole for the punch in the center was used, as shown in Figure 3, which ensured the specimen was fixed tightly.

Additionally, a dynamic load cell (Kistler 9132B, Kistler Japan, Tokyo, Japan), oscilloscope (Tektronix, DPO2014), and data recorder (OMRON ZR-RX70, OMRON, Kyoto, Japan) formed a data collection module to measure the dynamic force, as shown in Figure 4. Because of the high rigidity of the dynamic load cell (Kistler 9132B, Kistler Japan, Tokyo, Japan), it is suitable for measuring a rapidly changing force, and its reliability could be proved by the waveforms from the oscilloscope at 60 kHz, which is equal to the frequency generated by the ultrasonic generator.

### 2.2. Specimens and Forging Test Procedure

Commercial copper was selected to produce forging test samples, which were machined to a thin chip with a length, width, and height of 10 mm, 10 mm, and 0.2 mm, respectively. Since all the small pieces of chips were cut from a large one, the initial surface roughness was Rs = 220 nm.

To obtain the impact effect under a relatively low amplitude, the initial pre-load was aways set to 100 N in each forming test, since a higher pre-load would make it harder for the punch to depart from the surface of the material (Bai and Yang 2014) [15]; this, in turn, would make it more difficult to observe the impact effect. The vibration remained steady at 10 s in every processing procedure, because at the same frequency, plastic deformation only occurs in the initial 2 s; after this, the deformation is very limited (Yang and Bai, 2014) [16].

Under this pre-load, the critical amplitude for the occurrence of the impact effect will be produced between 1.5 µm and 2 µm, according to the waveform on the oscilloscope. To observe the effect of acoustic softening, low amplitudes of 0.5, 1, and 1.5 μm were chosen. To investigate at what stage the impact effect appears, amplitudes of 2, 2.5, and 3 μm were chosen. Higher amplitudes of 3.5, 4, 5, and 6 μm were selected to observe how the surface changes under a larger impact effect. The frequency in this study was always set at 60 kHz.

## 3. Results and Discussion

### 3.1. Separation of Acoustic Effect and Impact Effect

According to the explanation of the impact effect, the judgment standard for the occurrence of the impact effect is whether the punch can be observed to periodically contact and detach from the surface of the specimen during the process (Bai and Yang, 2014) [11]. In the case of a forging test, when a pre-load is loaded, a small amount of elastic deformation would be created on the surface of the copper chip. At that moment, the punch stays at the initial position. Then, as the vibration starts, the punch would move between the peak position and the valley position, the latter being the lowest position the punch can reach. The vibration amplitude is defined from the peak position to the valley position, as shown in Figure 5. At the low amplitude, if the created deformation by the punch stays at the valley position, it is not sufficient to cause plastic deformation; the elastic spring back would cause the material of the surface to recover its original state following the punch, when it moves from the valley position to the peak position and finishes a period of vibration.

As a result, the punch would contact the surface of the specimen during the whole process, which means the impact effect cannot occur in this situation. Moreover, the elastic force would be greater when the punch stays at the valley position and reduce gradually when the punch moves from the valley position to the peak position, so the dynamic force would be captured by the dynamic load cell and converted to an electrical input signal, which would then be displayed as an ideal sine wave on the oscilloscope. Figure 6a shows the waveform on the oscilloscope under a 1 μm amplitude.

In the contrary situation, with the amplitude increasing, there is the appearance of distortion on the waveform, as shown in Figure 6b. This is mainly due to the amplitude continuing to increase, and it makes the displacement of the punch deeper. Some parts of the asperity on the surface, especially in the bigger sizes, start plastic deformation, leading to the elastic force suddenly reducing while the punch is moving back to the peak position. A tiny slope then appears on the waveform. Furthermore, if the amplitude continues to increase, the slope of the distortion part on the waveform tends to flatten, as shown in Figure 6c, which indicates that more plastic deformation occurred on the surface asperity. When the amplitude reaches 6 μm, which is the maximum of the ultrasonic vibration, a straight line appears. This indicates that a majority of the asperities on the surface have commenced plastic deformation, and only a few still retain the ability to spring back. The dynamic load cell can hardly detect the elastic force.

According to the waveform displayed on the oscilloscope, the acoustic effect and impact effect could be separated successfully in this study. At the amplitudes of 0.5, 1, and 1.5 μm, only acoustic effects were seen to be impacting the process. After the amplitude was increased to over 1.5 μm, the impact effect appeared in the process, and with increasing amplitude, the impact effect was more significant.

### 3.2. Surface Finishing by Acoustic Effect and Impact Effect

To investigate which acoustic and dynamic effects are more significant in the micro-forging test, the surface topographies of copper chips were detected by an atomic force microscope (Keyence VN-8010). Three groups including ten samples in each group were processed by the forming process under the same conditions. Five points, on the top, bottom, left, right, and center, were selected on each sample’s surface. The most central position of each surface was used for topographic scanning for the observation of deformation on surface asperities, as shown in Figure 7.

The surface topographies of the original sample’s surface, and one of the forged sample’s surface, under different amplitudes of 0.5, 1, 1.5, 2, 3.5, and 6 μm chosen from these three groups are shown in Figure 8a–g, respectively.

The original surface roughness (Rs) of 220 nm was obtained before the forging test, and the maximum height of the asperity reached about 5956.5 nm, as shown in Figure 8a. After being processed with an amplitude of 0.5 μm, the highest asperity decreased to 2419 nm. However, with the amplitude increasing to 1 or 1.5 μm, which means the impact effect has not started yet, there was no noticeable reduction in the asperity. At this stage, the main deformation could be created by the pre-load from the punch, and only a small quantity of the asperity’s deformation is from vibration. Hence, the height of the asperity could still reach about 2666.89 nm and 2561 nm under amplitudes of 1 and 1.5 μm, respectively, as shown in Figure 8c,d. After increasing the amplitude to 2 μm, which suggests that the impact effect would be induced, a further reduction in the height of the asperity was caused. As shown in Figure 8e, the highest asperity is only 1784 nm. When the amplitude increased sequentially to 3.5 μm and then to 6 μm, the highest asperity also reduced to 1420 nm and 1234 nm, as shown in Figure 8f,g. This may be due to the greater plastic deformation created by the impact effect. The results imply that ultrasonic vibration is useful for improving the surface quantity. However, the surface’s asperity deformation is basically created by the impact effect rather than by the press machine, and acoustic softening has a very limited effect on the surface finish. Additionally, the improvement in the surface quantity of deformation of the asperity accumulates gradually and also increases with a large amplitude of ultrasonic vibration.

Figure 9 shows a 2D profile extracted from a 3D area along the red line in Figure 9a,b. Figure 9c shows the asperity formation after processing under a 0.5 µm amplitude; the asperity is presented as huge, sharp, and steep. However, after the impact effect occurred, as seen in the 3D topographies, the mountain became rounded, and some slopes were created, as shown in Figure 9d. Many places became flattened because of the impact, and the peak of the mountain was flattened; flattening even occurred in the valleys.

### 3.3. Effect on Reduction in Surface Roughness

In order to study the reduction in the surface roughness caused by acoustic softening and the impact effect, the ratio of the reduction in the surface roughness (ΔRa) was introduced, which can be calculated by following Equation (1):(1)$$\Delta R=\frac{R_{0}-R_{a}}{R_{0}}\times 100\%$$
where $R_{0}$ and $R_{a}$ are the surface roughness after the process and before the process.

Three groups were processed by the forming process under the same conditions. Every group included ten samples processed by different amplitudes of 0.5~6 μm, and the average values of surface roughness were calculated at five places on every sample, measuring about 200 μm^2^. They were measured by AFM, as shown in Figure 10 and Figure 11. These figures show the relationship between the reduction in the surface roughness and the different amplitudes. The surface roughness was also reduced after ultrasonic vibration started. However, only a small change occurred with acoustic softening, which highly conforms to a change in the surface asperity. The reduction in the surface roughness increased after the impact effect began. Although the reduction was not significant at the initial stage of the impact effect, a more obvious reduction could be obtained. The impact effect also became more intense when the amplitude was increased, and, finally, a roughly 50% reduction in the surface roughness was created by the impact effect.

### 3.4. Effect on Microstructure after Ultrasonic Forging

To understand how ultrasonic vibration influences the surface comprehensively, it is necessary to use EBSD to characterize the microstructural evolution. The results from EBSD indicate that the differences produced by acoustic softening and the impact effect at the microstructural level can be illustrated more clearly using this method, and it could be instrumental in helping us learn the mechanism of ultrasonic vibration on the surface.

Based on the above results, acoustic softening happened at amplitudes under 2 μm, and the impact effect commenced after 2 μm and continued until 6 μm was reached. Hence, samples with ultrasonic amplitudes of 0.5 and 1.5 μm were chosen to learn about acoustic softening; samples with ultrasonic amplitudes of 2 μm and 6μm were chosen to study the impact effect.

The selected samples were firstly cut from the centerline, and then two places were chosen to be observed. For comparing the difference in the ultrasonic vibration’s influence on the surface and entire bulk, one place close to the surface and another deeper one up to half of the height of the sample were chosen, as shown in Figure 12. Because the original samples were not annealed, the grain size in the surface is larger than in the inner area at the initial condition.

Comparing Figure 13a, which shows the EBSD maps of the grain orientation in measurement area 1, with Figure 13b, which shows measurement area 2, it is clear to see that the grain size gradually decreased with increasing amplitude. The more colorful grain in area 1 also indicates more grain rotation. However, the same trend was not observed in the center area, indicating that the influence of ultrasonic vibration has a limit. Meanwhile, Figure 13a shows that the reduction in the average grain size was more evident at amplitudes from 0.5 to 1.5 μm, which means acoustic softening was impacting the process. The grain size reduced from 13.2 μm to 9.3 μm. In addition, when the amplitude was over 2 μm, the trend of the reduction in the average grain size tended to flatten. Even when the maximum amplitude of 6 μm was reached, the average grain size was only reduced by about 1 μm, from 9.5 μm to 8.6 μm. Although grains and sub-grains could be distinguished when the average grain size decreased due to surface deformation, the reduction in the average grain size was only minimal under a low amplitude, and evident shear of the grains could not be found in area 1. However, comparing the grain size and form of the grains under an amplitude of 6μm to the original sample, the change in the average grain size was not obvious, but the form of the grains changed significantly. This may be caused by the shearing deformation and recrystallization induced by the local heating effect.

In order to observe the surface more directly, the larger version of the EBSD mapping on the surface, at about 5 μm depth, was carried out, as shown in Figure 14. Before processing, the surface was classified into three layers according to different grain sizes: the grain size in the upper layer is smaller than the bottom layer, as shown in Figure 14a. After processing under a 1.5 μm amplitude, the thickness of layer 2 became thinner, and most grains in layer 2 were refined to the same size as those in layer 1. Only a few residual grains did not change in size, which suggests that acoustic softening appeared during vibration, as shown in Figure 14b. The results after increasing the amplitude of the ultrasonic vibration to 6 μm are shown in Figure 14c. Note that it is entirely different from Figure 14b. Not only has layer 1 almost disappeared, but also the size of the grains in layer 2 is oppositely elongated. Additionally, the identical color of the grains indicates that the grains lie in one direction.

During the vibration from acoustic softening, energy was offered to the lattice defects, and it caused the defects such as voids or grain boundaries to create more dislocation; this created more rotation and more refined grains. However, in this process, the energy was consumed by the defect close to the surface and gradually disappeared before it reached the center material; as a result, the influence on area 2 was not evident. As the motion of dislocation was enhanced by acoustic softening, the capacity for plastic deformation also improved, which resulted in the 50 N pre-load reading on the data recorder also reducing slightly to about 40 N. The same phenomenon was also presented by Hu and Yang (2017) [13], although it is still hard to determine if this reduction in stress in micro-forging was induced by acoustic softening or not. As for the impact effect, due to the amplitude increases, the influence of acoustic softening reduced. Thousands of instances of hammering in a short period caused hardening to occur in the surface microstructure. Hence, as shown in Figure 14c, the grains were elongated, and no grain refinement occurred. On the other hand, when the impact effect became more dominant, an increase in the surface temperature could be observed. In the future, it would be worthwhile to investigate what might be created by the heating effect at higher amplitudes. 

## 4. Conclusions

The aim of this study was to investigate how acoustic softening and the impact effect affect the deformation behavior of the surface asperity, and changes in the microstructure of the material surface and interior under the influence of acoustic softening and the impact effect. A novel micro-forging test, which can separate acoustic softening from the impact effect efficiently, was used, and copper chips were applied to surface finishing by ultrasonic vibration at different amplitudes. Based on the results, the conclusions are as follows:

The high rigidity of the dynamic load cell and oscilloscope could be used to determine whether the punch detached from the surface of the specimen during micro-forging.The reduction in the performance of ultrasonic vibration on the surface roughness is highly in line with the change in the surface asperity. The results show that the impact effect influences the surface roughness more significantly than acoustic softening. The reduction in the surface roughness created by the impact effect increases to over 60% when the impact becomes more dominant with increasing amplitude; meanwhile, the reduction induced by acoustic softening only reaches about 20%.Acoustic softening will cause more grain refinement on the surface of the material; however, as the amplitude increases, the impact effect, which creates a greater heating effect, would oppositely cause grain growth.Ultrasonic vibration has the greatest influence on the surface of the material at about 5 μm depth, and it would decrease progressively from the top to 50 μm depth.

## Figures and Tables

**Figure 1 materials-15-01907-f001:**
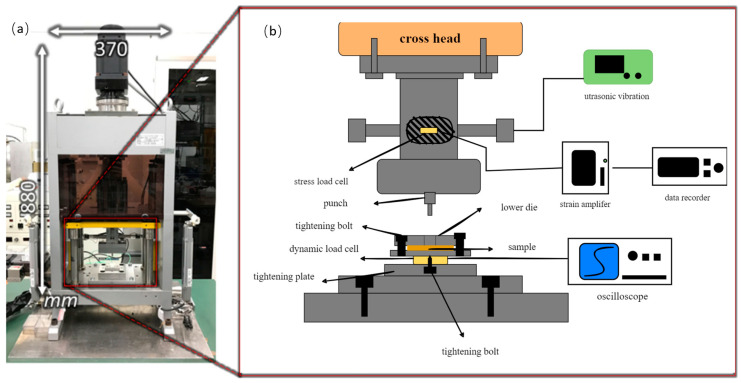
Configuration of the ultrasonic-assisted micro-forging system: (**a**) appearance of the desktop miniature servo press machine and schematic illustration of the micro-forging system. (**b**) Schematic illustration of the micro-forging system.

**Figure 2 materials-15-01907-f002:**
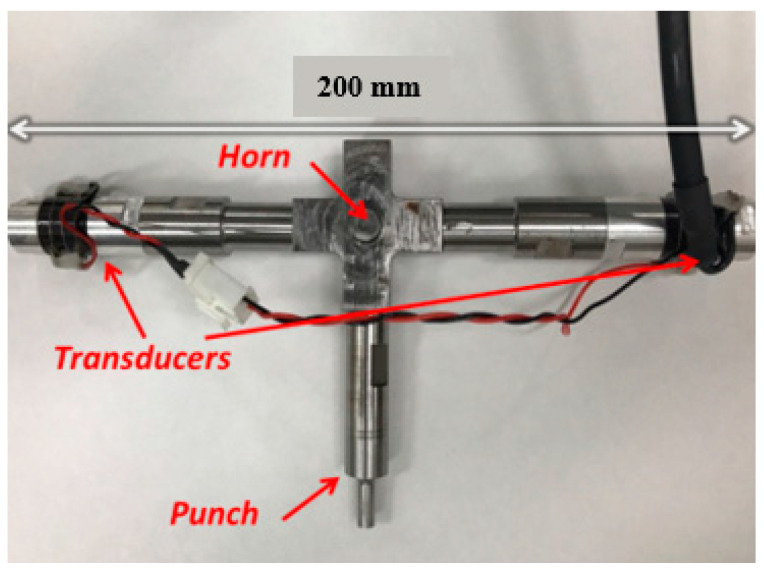
Integrated structure of the ultrasonic vibrator.

**Figure 3 materials-15-01907-f003:**
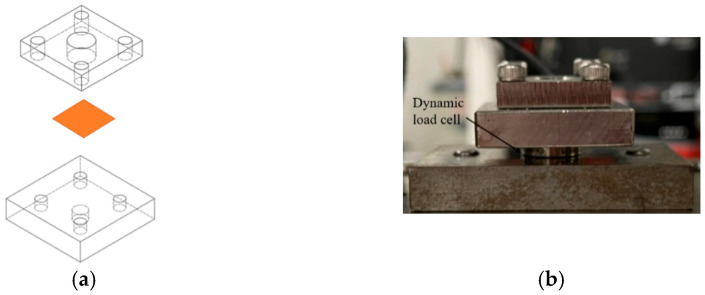
The fixation system of specimens: (**a**) schematic illustration of the fixation system; (**b**) photo of the fixation system.

**Figure 4 materials-15-01907-f004:**
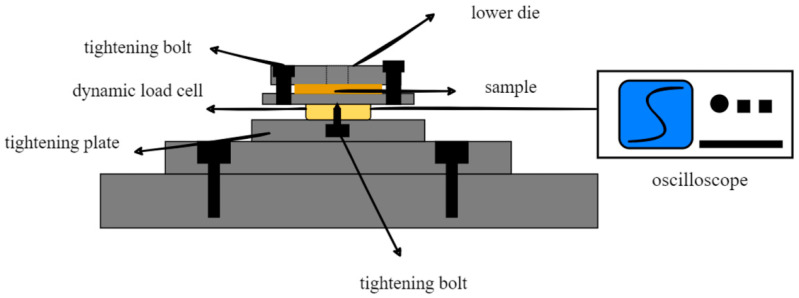
Dynamic force test system.

**Figure 5 materials-15-01907-f005:**
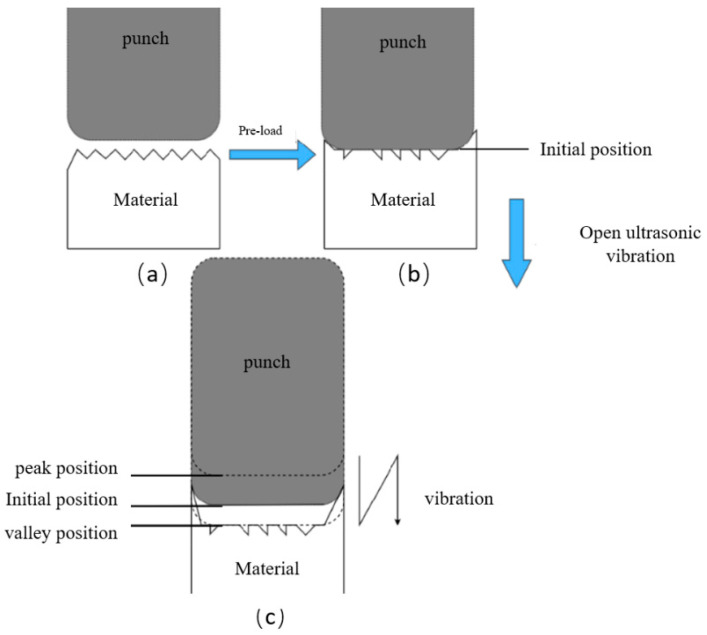
Micro-forging procedure of the punch. (**a**) no-load stage; (**b**) pre-load stage; (**c**) vibration started.

**Figure 6 materials-15-01907-f006:**
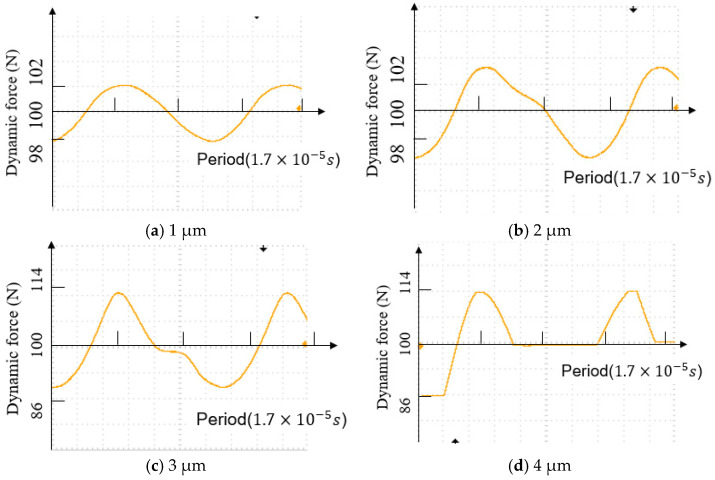
Waveform of dynamic oscillatory forces during compression with different ultrasonic amplitudes.

**Figure 7 materials-15-01907-f007:**
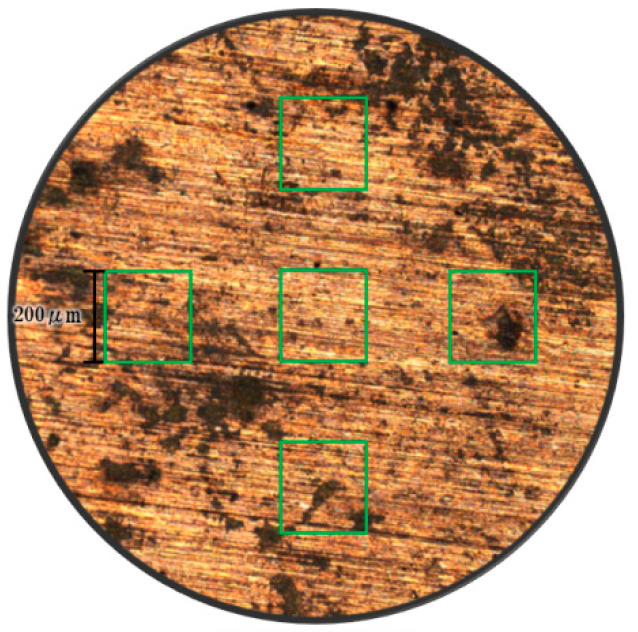
The top view of a sample’s surface showing the areas selected for surface analysis.

**Figure 8 materials-15-01907-f008:**
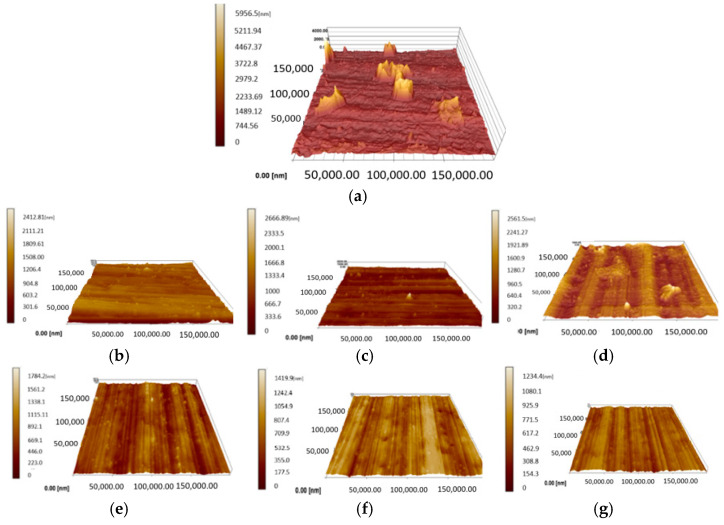
Three-dimensional surface topographies of the surface with different amplitudes: (**a**) original specimen, (**b**) 0.5 μm, (**c**) 1.00 μm, (**d**) 1.5 μm, (**e**) 2 μm, (**f**) 3.5 μm, (**g**) 6 μm.

**Figure 9 materials-15-01907-f009:**
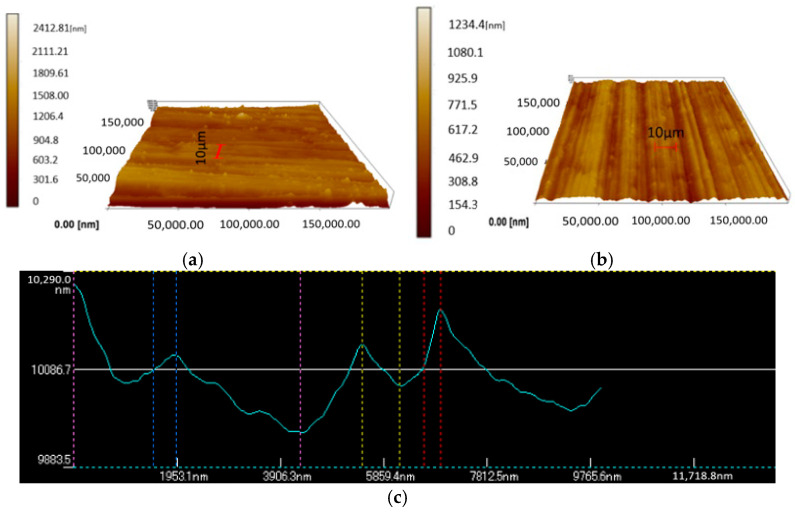
The 2D profile extracted from the 3D area. Topography of the sample at a (**a**) 0.5 μm amplitude and (**b**) 6 μm amplitude. The 2D profile of the sample at a (**c**) 0.5 μm amplitude and (**d**) 6 μm amplitude.

**Figure 10 materials-15-01907-f010:**
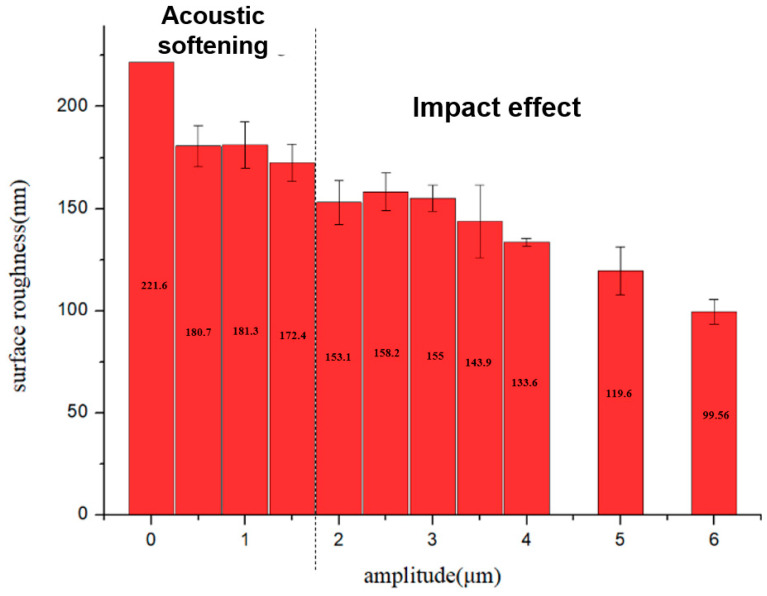
The average surface roughness at different amplitudes.

**Figure 11 materials-15-01907-f011:**
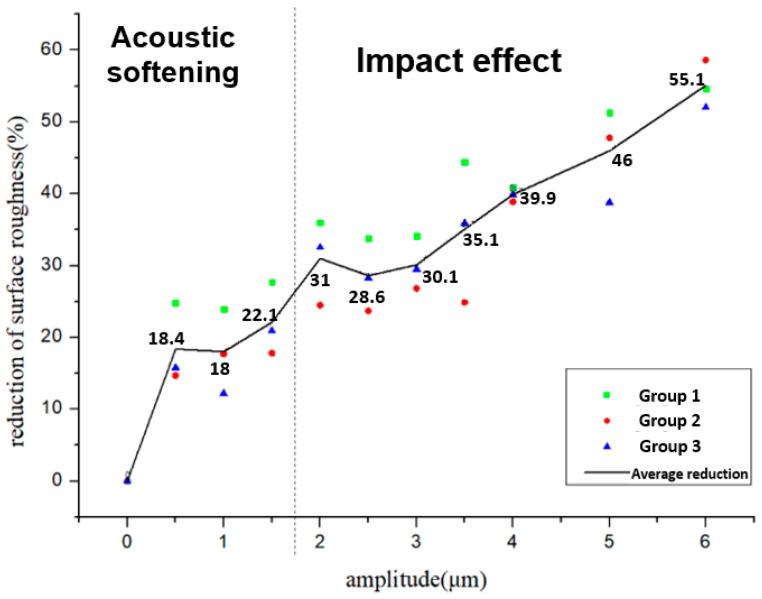
The reduction in the surface roughness–amplitude curve.

**Figure 12 materials-15-01907-f012:**
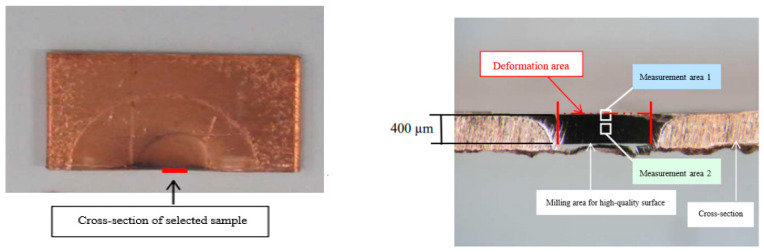
Schematic diagram of the EBSD mapping area.

**Figure 13 materials-15-01907-f013:**
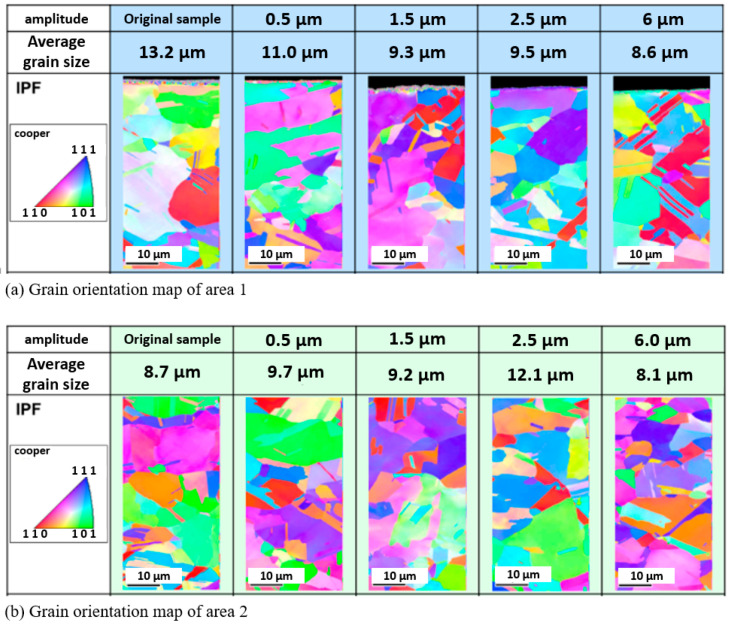
Result of the EBSD mapping in two areas.

**Figure 14 materials-15-01907-f014:**
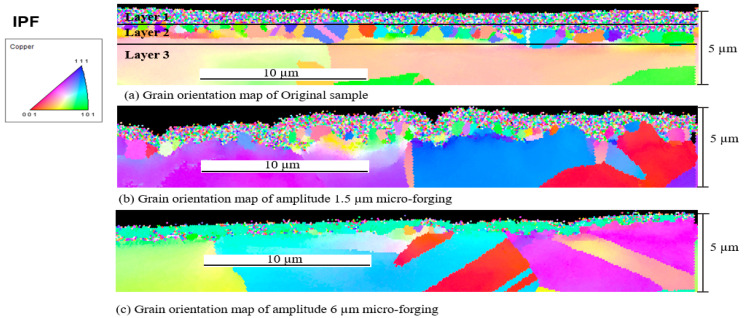
Grain orientation map at a surface scale of 5 μm.

## Data Availability

Not applicable.

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
