# Peer review of "Investigation on Deformation Behavior in the Surface of Metal Foil with Ultrasonic Vibration-Assisted Micro-Forging"

_materials, 2022, doi:10.3390/ma15051907_

Round 1
Reviewer 1 Report
Paper No.: Materials-1540142
Title: Investigation on deformation behavior in the surface of metal foil with Ultrasonic Vibration-assisted micro-forging.
This paper deals with a relatively new subject and applies an interesting novel testing methods and the achieved results are also valuable. However, there are some raised issues that need to be treated.
The English language needs to be reviewed.
Abstract
- The introducing part in (line 8 to line 15) is too long. Please reduce it to 2 or 3 lines.
- Usually abstract does not include references.
- The main finding results should be highlighted in the abstract.
Introduction
- “Nevertheless, some problems so-called size effect will be caused by scaling-down like the ratio of roughness to its dimension decreasing, increased forming stress, …” References are needed, such as: https://doi.org/10.1016/j.commatsci.2006.01.028 and https://doi.org/10.1016/j.tafmec.2006.05.002.
Experimental setup
- There is something wrong in the arrangement of the figures order, Fig. 2 appears at the beginning.
- In Fig.1 the upper and lower dies are not clear to identify. The arrow pointing to the punch is not in the right position.
- In the figure caption in line No : 106, please change “bibrator” to “vibrator”.
- Where is the location of the sample? Better include the sample in this setup.
- In line 114: “Punch surface will be re-ground to keep surface clearing and ensure ….” Please change to: “Punch surface will be re-ground to keep the surface clear to ensure …...”.
- Please re-write the sentence in lines No. 125-126.
- In line No. 140: the deformation is very limit …… please change “limit” to “limited”.
- Will the preload (100N) be vibrated by the mentioned 10 different amplitudes? Please clear it.
- In Fig. 4 please change “toghtenning bolt” to “tightening bolt”.
- In figure 5, please number it to a, b and c. in Fig 5-c, the dotted curve showing the valley is not at the valley of the asperities. If the authors mean by valley the lower position of the punch, please clear it.
Results
- The legends of Fig. 7 showing the scale of the topography needs to in higher font size to be seen.
- “Three groups of samples are processed by forming process …..”. What are the differences of these three groups? Do the authors mean that three samples for each condition were processed. Please make it clear.
- Caption of Fig. 11 is placed under Fig 12, and Fig.12 has no caption.
- Please explain the change of grain size of the original sample by changing the position from are (1) to area (2).
Conclusions:
- Conclusions No. 1&2 are long and can be formed as recommendations elsewhere.
- Conclusions No. 3 could contain the key results in numeric figure.
References:
The references format is not consistent.

Author Response
Dear Reviewers:
We are very grateful to Reviewer for reviewing the paper so carefully.
We have carefully considered the suggestion of Reviewer and make some changes.
A cover letter was attached as responsed to your comments.
should you have any questions, please contact us without hesitate.

Reviewer 2 Report
Dear authors, manuscript ‘Investigation on deformation behavior in the surface of metal foil with ultrasonic vibration-assisted micro-forging’ have much weakness that must be clarified and significantly improved. Please find below some suggestions:
- Is this suitable to cite papers in an ‘Abstract’ section? Usually, both template and journal requirements do not allow to proceed with references in the manner presented.
- Staying with an ‘Abstract’, it should be more general information about resolving the scientific problem than main conclusions from previously published papers. This section must(!) be modified radically.
- It was not accurately presented what is the ‘vibration amplitude’. Some introduction to this sentence might have been required. Is this defined by changes in surface roughness height or, respectively, other values? It is more my understanding in the further manuscript body than properly introduced at the beginning.
- What is the ‘FF modal analysis’? – line 103. Even very popular methods should be clarified.
- Bolded sentences (lines 96 and 130, respectively), should be divided by numbering, e.g. 2.1 and 2.2, subsections to make the reader a better understanding of the information that authors trying to convey.
- Except for many editorial mistakes, mentioned at the end of the review, there is in lines 143-146 unknown sentence ‘3. Results… This section may be divided…’. It looks extremely confusing like the authors did not remove their (own) comments. Moreover, Figure 5 looks like additional data that was missed to be removed by authors before manuscript submission. If the figure is referenced in the text, should be located after citation.
- Were there any surface topography measurement errors taken into consideration? Maybe, at least, uncertainty? The AFM measurement method, as any in surface metrology, can be fraught with many disturbances that influence the accuracy of the results obtained. Even highly-precise measuring equipment may not provide relevant results if the accuracy of a data processing method is lost. Please look for some examples in the reduction of data processing errors when a measurement errors, noise, in particular, is analysed:
(1) https://doi.org/10.1088/1361-6501/aab528
(2) http://dx.doi.org/10.24425/mms.2020.132772
(3) https://doi.org/10.3390/app7010054
- Maybe it seems like a popular suggestion, nevertheless, the Ra parameter responds more to the profile (2D) properties than an areal (3D). In fact, till today, surface roughness is often studied with a profile performance, especially, e.g. ground or turned, machined in general, surfaces (details). Therefore, why do not present both data, profile (2D) and areal (3D)? Some, various, examples would be nice to find by a reader.
- Can not find Figure 12, mentioned in lines 269-296. It appears hopelessly confusing and, correspondingly, gives an impression of being written quickly or, at least, carelessly.
- In my understanding, ‘a novel forging tests’, lines 298-299, were found (proposed) in the reviewed manuscript. From all of the manuscript studies, it looks like many of the results and, respectively, conclusions, have been provided with previous researches, presented in already published papers. This is appreciated if the current results are a continuation of previously conducted studies, nevertheless, the novelty of the manuscript presented should be highlighted.
- The ‘References’ section is in a heavy messy. There are much full information missed. Some papers are abbreviated, and some are not. It should be unified according to the journal (template) requirements.
Moreover, many editorial errors were found:
- Double-space in lines 28, 43, 62 (twice), 143, 176, 178, 179, 272 and 290, respectively.
- Falsely presented (formatted) references in text in lines 67, 78, 80, 154 and 289.
- Sentences should start with a capital (or a lowercase) letter in lines 75, 96, 127, 136, 169, 172, 208, 231, 234, 268 and 302.
- The sentence ‘…were carried out at and ten specimens…’, lines 141-142, should be improved.
- Some Figures' references in the text are not properly written, e.g. line 172 ‘Fig .6(b)’ or line 177 ‘Fig .6(c)’.
- There is a dot missing at the end of line 218.
Generally, the proposed manuscript in some parts presents messy. In some moments, it was difficult to follow the authors’ proposals. Some issues make an understanding of the paper difficult and the reader confused. Many editorial mistakes makes the paper even weak. Therefore, the manuscript should be significantly improved before any further consideration.
Author Response
Dear Reviewers:
We are very grateful to Reviewer for reviewing the paper so carefully.
We have tried our best to improve the manuscript and have modified the problem you mentioned.
A cover letter was attached as responsed to your comments.
should you have any questions, please contact us without hesitate.

Round 2
Reviewer 1 Report
No more comments.
Author Response
Dear Reviewers:
We are very grateful to Reviewer for reviewing the paper so carefully.
should you have any questions, please contact us without hesitate.
Reviewer 2 Report
Dear authors, your manuscript, Manuscript ID: materials-1540142, even revised, have some flawless that makes him weak. Please read below:
- Abstract even improved, seems to be weak and makes the reader not interested in further proceeding with the paper. Two sentences from lines 15-17 did not allow to properly attract the regular reader what can be received after 16 pages of study.
- The ‘Introduction’ section, in fact, contain much valuable information about the topic raised. Nevertheless, a critical review of the literature was not provided. Hence, the novelty was not justified in a required manner.
- According to comment no 3 in the first review, there was not justified how the amplitude value of vibration was selected. In sentences from lines 153-162, there was messy and proposed 10 amplitude were, at least in written form, not supported, even some paper(s) was references. Moreover, Bai Yang 2014 (line 156) and Yang Bai, 2014 (line 159) look poor.
- There is no word about the accuracy of the surface topography measurement or, respectively, uncertainty. At least, usually, even measurement errors are not considered, the effect of uncertainty is reduced with repeating the measurement of the samples. There was no word about that, even crucial against the whole analysis, matter. Proposing interesting methodology does not provide valuable results when received data may not be classified as relevant. From that point of view, bot all the studies and accuracy improvement may be lost.
- When previous suggestion no 8 was introduced it was required to specify which profiles were analysed. The direction of the profile (2D) extraction from the areal (3D) data can define the amplitude of the profile. It was not mentioned if profiles were selected with a vertical or horizontal (or maybe other) performance. Therefore all of the studies seem to be unsupported or, being more precise, unsubstantiated.
- The ‘conclusion’ section was modified, however, it seems to be short and does not present the novelty, which is, presented in this form, lost. It was not highlighted that, usually, the most valuable conclusion is provided firstly, when some other, supporting that the most useful. From this section, it is difficult to follow the novelty.
- Moreover, there are still difficult for the regular reader, to define what is the current novelty of the studies provided, when much similar analysis was received in previous research of the author.
Generally, it was found very confusing that many issues, even those mentioned in the first review, were addressed too shallow and, unfortunately, make the reader difficult to follow what the authors are trying to convey. From all the above, the revised manuscript still seems to be weak.
Author Response

(The authors gave the same response as above.)
